# Data-Driven Deep Journalism to Discover Age Dynamics in Multi-Generational Labour Markets from LinkedIn Media

**Abeer Abdullah Alaql [1], Fahad AlQurashi [1] and Rashid Mehmood [2,*]**

1  Department of Computer Science, FCIT, King Abdulaziz University, Jeddah 21589, Saudi Arabia
2  High-Performance Computing Center, King Abdulaziz University, Jeddah 21589, Saudi Arabia
*  Correspondence: rmehmood@kau.edu.sa

**Abstract:** We live in the information age and, ironically, meeting the core function of journalism—i.e., to provide people with access to unbiased information—has never been more difficult. This paper explores deep journalism, our data-driven Artificial Intelligence (AI) based journalism approach to study how the LinkedIn media could be useful for journalism. Specifically, we apply our deep journalism approach to LinkedIn to automatically extract and analyse big data to provide the public with information about labour markets; people's skills and education; and businesses and industries from multi-generational perspectives. The Great Resignation and Quiet Quitting phenomena coupled with rapidly changing generational attitudes are bringing unprecedented and uncertain changes to labour markets and our economies and societies, and hence the need for journalistic investigations into these topics is highly significant. We combine big data and machine learning to create a whole machine learning pipeline and a software tool for journalism that allows discovering parameters for age dynamics in labour markets using LinkedIn data. We collect a total of 57,000 posts from LinkedIn and use it to discover 15 parameters by Latent Dirichlet Allocation algorithm (LDA) and group them into 5 macro-parameters, namely Generations-Specific Issues, Skills and Qualifications, Employment Sectors, Consumer Industries, and Employment Issues. The journalism approach used in this paper can automatically discover and make objective, cross-sectional, and multi-perspective information available to all. It can bring rigour to journalism by making it easy to generate information using machine learning, and can make tools and information available so that anyone can uncover information about matters of public importance. This work is novel since no earlier work has reported such an approach and tool and leveraged it to use LinkedIn media for journalism and to discover multigenerational perspectives (parameters) for age dynamics in labour markets. The approach could be extended with additional AI tools and other media.

**Keywords:** media; journalism; deep journalism; labour markets; great resignation; quiet quitting; millennials; Generation Z; big data analytics; Natural Language Processing (NLP)

## 1. Introduction

### 1.1. The State of Journalism

We live in the information age and, ironically, meeting the core function of journalism—i.e., to provide people access to unbiased information—has never been more difficult. Perhaps the reason lies in the fact that people were never so distant from their real, physical, environments as they do now. This seems obvious given that we are connected to the virtual world that is pulled in front of our eyes by people whose intent may not be known or the intent may be to provide information that serves the business or the powerful. A perspective on this was provided by Herman and Chomsky in their book "Manufacturing Consent: The Political Economy of the Mass Media" (Herman and Chomsky 1988). They define an analytical framework called the "propaganda model" and note, "A propaganda model focuses on this inequality of wealth and power and its multilevel effects on mass-media interests and choices. It traces the routes by which money and power are able

to filter out the news fit to print, marginalize dissent, and allow the government and dominant private interests to get their messages across to the public". They further note that, normally, top-down intervention is not used to keep journalists under control, rather by "journalists' internalization of priorities and definitions of newsworthiness that conform to the institution's policy" (Herman and Chomsky 1988).

Another reason for the challenges facing journalism is the increasing complexity of the world and information, requiring new methods for journalism. Phenomena such as populism, partisanship, and kleptocracy have led societies to extremism (Goodwin 2012; Alvares and Dahlgren 2016). Consequently, public trust in governments has plummeted dramatically. The main issues are related to the perception of the public that it is the responsibility of others, not themselves, to provide impartial information and ideal or acceptable governance. The American Press Institute defines journalism's purpose as "to provide citizens with the information they need to make the best possible decisions about their lives, their communities, their societies, and their governments" (American Press Institute 2022). Traditional journalism has failed to achieve this goal for a variety of reasons, including difficulties in maintaining media organisations' freedom and impartiality, as well as funding cuts, which have led to public distrust (Carlson 2019; Allern and Pollack 2019). This is highlighted by UN News, which quotes UN Secretary-General António Guterres as saying, "at a time when disinformation and mistrust of the news media are growing, a free press is essential for peace, justice, sustainable development, and human rights" (UN News 2019).

The general public's distrust of traditional journalism, combined with the rise of the Internet, digital technologies, and digital and social media, has resulted in a rapid rise of citizen journalism, i.e., journalism by the general public, particularly through social and other digital media (Wall 2015; Raza et al. 2022; Zeng et al. 2019; Carmichael et al. 2019; Sienkiewicz 2014). Participatory journalism, democratic journalism, public journalism, and other terms were used to describe citizen journalism. Citizen journalism is difficult to understand and govern due to its very nature–i.e., it should be free—and because it is multifaceted, multidimensional, multilevel, and multimodal (Nah and Chung 2020). Citizen journalism solves some of traditional journalism's problems. However, it also introduces new ones, for example, subjectivity and, potentially, could lack responsibility, quality, standards, and regulations (Al-Ghazzi 2014; UN News 2022; Mutsvairo and Salgado 2022). Everyone bears the responsibility for upholding ideals, so everyone is accountable and must work to uphold freedom, equity, sincerity, honesty, and other ideals. Traditional and citizen journalism must collaborate, and their issues must be rectified cohesively by the people.

### 1.2. Labour Markets and Multi-Generation Era

We are affected by economic issues every day, such as inflation, energy, taxes, and interest rates. The global COVID-19 pandemic, environmental disasters, geopolitical situations, and wars have exacerbated the economic and social situations throughout the world, around developed and developing countries, and everywhere else. Experts, the media, politicians, and ordinary citizens are all concerned about the national and global economic situation these days. For example, Braun notes that policies of the Federal Reserve, the continuation of the COVID-19 pandemic (new variants, vaccination, and their effects), labour shortages, supply chain vulnerabilities, and US-China relations are among the top economic risks this year (Braun 2021). Kalish reported an update on the global economy for August 2022 and listed several economic concerns including global inflation, low consumer spending, effects of climate change on global economies, increase in labour costs, gas supplies for Europe, and global food security (Kalish 2022).

Moreover, our societies and workplaces are becoming more multipolar, rather than multilateral, with increased segmentation along generations, genders, races, ideologies, and several other dimensions. People are polarised and hate each other, even based on football clubs. Societies were never like this, not so complex, and not so divided.

Another current major economic issue is the Great Attrition (also called Great Resignation or Big Quit) accompanied by Quiet Quitting (Ellis and Yang 2022). People are resigning from their jobs in great numbers around the world. For example, over 24 million Americans left their jobs during the second and third quarters of 2021 (Sull et al. 2022). Based on an extensive study of over 30,000 people, Microsoft reported that more than 40% of the global workforce is considering leaving their employers in 2021 for reasons such as the need for flexible and hybrid work, the gaps between business leaders and employees, high productivity requirements causing exhaustion and burnouts in the workforce (meetings, emails, chats, etc.), Gen Z (particularly) struggling with work-life balance, and others (Microsoft 2021). According to the World Economic Forum, the Great Resignation is not over, and a fifth of workers on average plan to quit in 2022, with as many as 66% of unhappy workers in India planning to leave within the next 3–6 months, followed by Singapore, where 49% of workers plan to leave their jobs (World Economic Forum 2022). The study found that compensation, meaning (self-fulfilment), confidence or competence (allowing creativity), and autonomy (time, place, and flexibility) were the key factors when considering a job change. Quite Quitting has aggravated the problems of Great Resignation. It is seen differently by different people. Some people see Quiet Quitting as a movement to do the minimum at work or untethering careers from identities, while others see it as a reminder to not work to the point of burnout (Ellis and Yang 2022).

The worsening and uncertain situation with the economies and labour markets globally calls for increased research in this area. For instance, research in labour markets is needed to understand the evolving nature of labour (skills, education, generation-specific trends, and preferences, etc.) and employers (industry trends, priorities, ESG (Environment, Social, and Governance), compliance to various regulations, and Triple Bottom Line (TBL), etc.). Research in labour markets is also needed to understand the role of evolving societies in labour markets and comply with national and global priorities such as UN Sustainable Development Goals (SDGs). Research in these areas is urgently needed to develop new labour economics and labour markets for sustainable economies and societies. Additionally, understanding various generations in workplaces or societies (silent generation, baby boomers, generation X, millennials, generation Z, and generation alpha) is very important in the current climate.

### 1.3. LinkedIn: A Trusted Professional and Business Media

Online Social Networks have continued to grow in number, quality, and significance, and with this, their influence on society is becoming more profound (Ahn et al. 2007; Tobi et al. 2013; Mislove et al. 2007). This is especially true due to the emergence of smartphones, wearables, and other smart mobile devices. This cyberspace data could be used to capture public opinion; identify communities and organisations; detect trends; obtain predictions in any area where sufficient data is available; and investigate any topics of interest, the possibilities are endless (Suma et al. 2020; Alotaibi et al. 2020; Alomari et al. 2021).

LinkedIn has 850 million subscribers and is regarded as the most popular social networking site for professionals (LinkedIn Corporation 2022). It provides users with the opportunity to display their academic achievements and form connections. LinkedIn is a key media being used for recruitment, content distribution, and building brands. It is possible to capture issues and detect trends based on the vast amount of valuable information from LinkedIn posts. We used LinkedIn media because it offers massive repositories of social media data between professionals.

The paper is organised as follows. Section 2 provides the literature review, briefly describes our work, and establishes the research gap and novelty of the work presented in this paper. In Section 3, we describe our approach, the methodology and the tool design. The implementation is discussed in Section 4. The discovered parameters are discussed in Section 5. A discussion is presented in Section 6. Conclusions and future directions are presented in Section 7.

## 2. Literature Review and Description of Our Work

This section discusses the works related to the research presented in this paper (Section 2.1), provides a brief description of our work (Section 2.2), and summarises the research gap and novelty of our work (Section 2.3). A comprehensive review of academic research on using artificial intelligence (AI) and data analytics for LinkedIn was conducted. We did not find any work similar to our paper. We review works within the context of data analytics using LinkedIn profiles or other related social media.

### 2.1. Related Works

Social media data mining is a process that involves collecting and analysing social media data. Data mining is a type of data analysis that uses machine learning and other advanced technologies to find patterns in social media data. For example, it can be used to gain knowledge about the demographics of its users (Gjurković et al. 2021). In addition, it can help companies to identify employees' skills and qualifications in the workplace. For instance, using LinkedIn as the basis for user modeling, Alruwaili and Alahmadi (2021) extracted LinkedIn profiles. Then, LinkedIn users were clustered based on their skills. Furthermore, Dai et al. (2015) scraped public profiles using a scraping technique. Then, NLP techniques are applied to the classification of educational background and the clustering of professional background in collected profiles.

Purwono and Wulandari (2021) used LinkedIn profile photos to predict face shapes. In addition, using publicly available data, Nguyen et al. (2021) explored the LinkedIn profiles of Australians. Based on facial feature extraction and analysis, K-means clustering, and Principal Component Analysis were found to be viable techniques for classification of users.

Kumalasari and Susanto (2020) used the K-Means Clustering algorithm to test whether using IT professionals' data as a reference is feasible. To classify students based on their skills and the information technology field, the Collaborative Filtering method by the K-NN algorithm is employed. Based on the collected IT student skills, the recommendation for the IT job field is generated. Furthermore, Dai et al. (2018) used Clustering methods to analyse LinkedIn profiles based on their professional backgrounds. The authors analysed the trending professional orientations of the workforce. Moreover, Domeniconi et al. (2016) used the data from LinkedIn users' public profiles to identify relationships between jobs and people skills. Giri et al. (2016) developed a hassle-free automated process that enables recruiters to select candidates who fit their organisation. Using Greedy, Hierarchical, and K Mean clustering algorithms, Garg et al. (2016) clustered LinkedIn profiles by job title, company name, and geography. On the other hand, in order to create domain-specific job understanding models, Li et al. (2020) applied deep transfer learning.

Researchers have also used other types of social media data such as Twitter data to automatically extract information using machine learning in various research fields. For example, using Twitter data, Haghighi et al. (2018) proposed a paradigm for evaluating transit riders' perceptions of service quality. For filtering tweets that are related to the real customer experiences of the transit system, the authors employ topic modeling by using an unsupervised machine learning approach. Sentiment analysis is then conducted utilising an established tweet-per-topic index to evaluate transit riders' opinions as well as investigate the root causes of their unhappiness with the service. This methodology has the potential to be very helpful to transportation authorities in terms of user-oriented analysis and investment decision-making. Moreover, Alomari et al. developed an automatic labeling approach (Alomari et al. 2021) for machine-learning-based traffic-related event identification using Twitter data in Arabic. The implemented software programme is named Iktishaf+ that uses distributed smachine learning over Spark. In addition, Karami et al. (2020) discussed topics in Twitter-based studies, to summarise the temporal trend of those topics, and to interpret their evolution over the last decade. This study utilises a productive strategy to mine many Twitter-based studies to characterise the relevant studies. Moreover, Martínez-Rojas et al. (2018) employed a research methodology based on

a systematic review of the literature to provide an overview of the current state of research on the use of Twitter in emergency management, and to identify problems and potential suggestions for further studies.

*2.2. This Work*

Motivated by the challenges in the journalism and labour market fields, in this paper, we explore deep journalism, our data-driven Artificial Intelligence (AI) based journalism approach to study how the LinkedIn media could be useful for journalism. Specifically, we apply our deep journalism approach to LinkedIn to automatically extract and analyse big data to provide the public with information about labour markets, people's skills and education, and businesses and industries from multi-generational perspectives. We combine big data and machine learning to create a whole machine learning pipeline and a software tool for journalism that allows discovering parameters for age dynamics in labour markets using LinkedIn data.

We collect a total of 57,000 posts from LinkedIn and use it to discover 15 parameters by the Latent Dirichlet Allocation algorithm (LDA) and group them into 5 macro-parameters. These macro-parameters represent five different views on multi-generational labour markets. (i) Generations-Specific Issues that include the relationship between unemployed young people and crimes, unemployment and drug abuse caused by family breakdown, etc., (ii) Skills and Qualifications macro-parameter includes leadership skills such as communication and effective decision making, nurturing leadership in the young generation, job-specific skills, training, and more. (iii) Employment Sectors include parameters such as changing workspaces, remote and hybrid work, recruitment sector, family businesses, entrepreneurship, and more. (iv) Consumer Industries include celebrations and events industries, the retirement industry, band marketing, energy sectors, and the entertainment industry. (v) Employment Issues include mental health and others. We also discover and elaborate on labour market-specific characteristics and preferences of the multi-generational workforce such as increased use of music consumption among Generation X and Y, a high likelihood of a criminal record among unemployed Generation Z, a focus on mental health among Generation Z, and others.

We implemented the proposed data-driven approach for age dynamics parameter discovery from labour markets data into a software tool. The tool comprises multiple software components that collect, process, discover parameters, analyse, visualize, and validate the discovered information. The details of the methodology and the software tool for parameter discovery including the data collection procedure and analysis of the parameters are provided in Sections 4–6.

The journalism approach used in this paper can make objective, cross-sectional, and multi-perspective information available to all. It can bring rigour to journalism by making it easy to generate information using machine and deep learning and can make tools and information available so that anyone can uncover information about matters of public importance. The work also contributes to an understanding of labour markets and that could be used to create awareness of important issues among the public, academics, and others, and drive future research on this topic using cutting-edge technologies. The parameters discovered and the knowledge gained could be used by the public to make informed decisions, as well as to direct labour economics research in important areas. It is expected that this work will contribute to developing the theory and practice of AI-based journalism, the use of LinkedIn media for journalism, and novel approaches toward labour economics and labour markets facilitated through the knowledge discovery and journalism approach, eventually leading to the development of sustainable societies and economies.

The work presented in this paper on journalism and media is a part of our wider work that aims to develop ICT technologies to solve pressing problems in smart cities and societies. For example, we introduced the concept of Deep Journalism in (Mehmood 2022; Ahmad et al. 2022) and discovered public, academic, and industry perspectives on transportation using The Guardian, Web of Science, and Traffic Technology International

Magazine, respectively. We have also used a similar methodology to discover parameters for smart families and homes (Alqahtani et al. 2022), healthcare services for cancer (Alahmari et al. 2022), and education and learning during the COVID-19 pandemic (Alswedani et al. 2022). Earlier we developed a range of works on AI-based event detection, such as Alomari et al. (2020).

### 2.3. Research Gap and Novelty

The literature review provided earlier in this section shows that the earlier work on machine learning-based analysis of LinkedIn data focused primarily on users' profiles aiming to understand people's skills and jobs and their interrelationships. None of the earlier works explored an AI-based approach to studying how LinkedIn media could be useful for journalism. None of the works has applied a journalism approach to LinkedIn to automatically extract and analyse big data to provide the public with information about labour markets; people's skills and education; and businesses and industries from multi-generational perspectives. The (deep) journalism approach and the use of Linked media for journalism, the knowledge discovered on age dynamics in labour markets, the methodology of discovering parameters for labour markets, the extracted taxonomy, and the quantitative and descriptive analysis of age dynamics in labour markets, all these are novel contributions of our work. The approach could be extended with additional AI tools and other data sources.

### 3. Methodology

The purpose of this section is to explain the methodology and the design of the proposed architecture of the system. Figure 1 shows our software architecture, which is made up of four components, each of which will be discussed further in the following section.

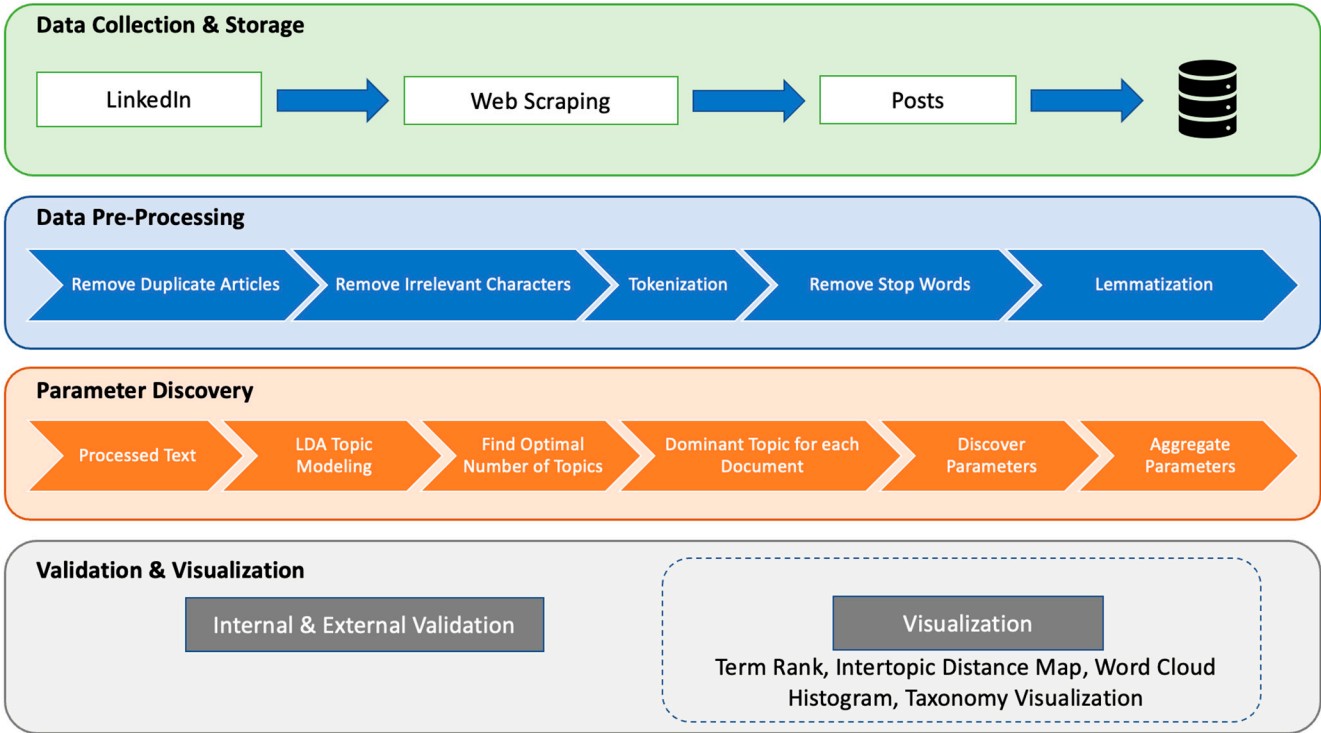

**Figure 1.** The System Architecture.

We first use a Web scraping technique to collect LinkedIn posts. Using a specified search query term, we collected the posts for our dataset. Post objects were collected as JSON (JavaScript Object Notation) objects. Then, we converted the JSON files into a CSV file. Afterwards, the CSV file is loaded and preprocessed with Pandas. We then

used LDA-based parameter discovery module by Gensim Python library (Řehůřek 2022). Using domain knowledge, we carried out quantitative analyses, categorised the clusters as parameters, and then grouped the parameters into macro-parameters. Finally, we visualised the parameters and macro-parameters. Additionally, internal, and external validation techniques were used to validate these parameters.

## 4. Implementation

The purpose of this section is to explain the implementation of the proposed architecture of the system. Sections 4.1–4.7 explain data collection, the sources of data we used in this study (LinkedIn); preprocessing; parameter modelling; parameter discovery and quantitative analysis; validation; and visualisation.

### 4.1. Data Collection

Since the multi-generational workforce is a topic related to multiple disciplines, an iterative process was utilised to determine the key search terms for the queries. An extensive literature review was conducted first to identify alternative synonyms used in the literature to refer to the multi-generational workforce. The final key terms used in the search query are listed in Table 1.

**Table 1.** Data Collection.

| Source | Query | Posts |
|---|---|---|
| LinkedIn | Millennials, Gen Y, baby boomers, Generation Y, Generation X, Gen X, Gen Z, Generation Z, Alpha generation, multi-generation, Inter-generational, cross-generational, Employment, Unemployment, Middle Aged, Aged, Elderly, Young, Senior, Seniors, 20s, 30s, 40s, 50s, 60s, 70s, 80s, 90s, Ageism, Labourforce, Demographics, Workforce, Workplace, Workers, Retirement, Recruit, Aging, Leadership, Entrepreneur, Preschool Child, Child, Preschool, Infant, Children. | 57 K |

### 4.2. Dataset: Posts (LinkedIn)

LinkedIn (LinkedIn Corporation 2022) is one of the most widely used professional networking and social media sites in existence today. With over 850 million users, LinkedIn has become the go-to place for job seekers, businesses, and professionals to make connections, find new opportunities, and build their personal brand.

Since technologies and systems are constantly evolving, and new generations are entering the workforce, multigeneration workforce is changing the ways in which companies and organisations operate. Thus, non-personal information in LinkedIn posts could help us to make better decisions, understand, and analyse multi-generational workforces.

In this study, we collected our dataset from LinkedIn (posts). Thus, the LinkedIn website was obtained using web scraping techniques including Python (2022), Beautifulsoup (2022), Requests (2022), and Pandas (2022). The data was collected for a period of four months (March 2022–July 2022) using the search query terms listed Table 1. These posts are anonymised (aggregated) data, and we do not use personal information. Then, posts are saved in a CSV file, including around 57K posts. The CSV file includes two columns, the query term, and the post.

### 4.3. Data Preprocessing

The pre-processing steps are as follows: remove duplicate articles, and irrelevant characters, tokenize stop words, and lemmatize with POS tags, among other things. Pandas is a Python package that is used as the first step in reading the CSV file and saving it in a data frame (DF). A second step consisted of removing all redundant data and the third step consisted of eliminating all unnecessary characters, such as several Unicode characters. A simple preprocess function from the Python package "Gensim" was used in the fourth step to tokenize the texts. The fifth step was to eliminate the article's stop words. Lemmatization was performed using Spacy (spaCy 2022) as the last step of data

pre-processing. As a result of preprocessing, the cleaned posts were obtained, which were then used in the LDA model.

### 4.4. Topic Modelling Using Unsupervised Machine Learning

In document analysis, topic modelling is a machine-learning technique for discovering topics from a set of documents (Wang et al. 2012). Topic modelling is used in various document analysis applications such as information filtering, retrieval, and semantic search. Topic modelling is also known as latent semantic analysis or latent topic analysis. This method learns the hidden themes in a body of text by analysing words and their variations appearing in different contexts.

We used Latent Dirichlet Analysis (LDA) (Maier et al. 2018), a topic modelling technique, in this research. The parameters of our LDA model were set to 15 topics, 10 passes, and 100 iterations. The number of topics is considered an essential model-building parameter. When the number of topics is great, the model must be overfitted, but when it is small, it must be under fitted. To determine the ideal topic model (optimal number of topics), we created many LDA models with varying topics (k) and chose the optimal number based on coherence measures and the visualisation of the LDA models. Passes refer to the number of times an algorithm must traverse the full corpus. The maximum number of iterations is required to determine the probability of each topic in the corpus.

### 4.5. Coherence Measures

Coherence scores (Röder et al. 2015) in the topic modelling are used to gauge the interpretability of a topic for humans. The topic coherence scores provide a comprehensive way to compare different topic models. It compares by capturing an optimal number of topics and assigning a number called 'Coherence Score' for the interpretability of those topics. Measuring the coherence of topics was studied recently to solve the issue of topic models not providing any guarantees about the interpretability of their output. The most widely used approaches for tuning the LDA hyperparameters are based on various topic coherence measures. In this study, we usedfour coherence measures: C_v measure, C_uci measure, C_umass measure, and C_npmi measure.

### 4.6. Parameter Discovery and Quantitative Analysis

We discover the parameters and macro-parameters using domain knowledge and quantitative analysis methods such as term scores, intertopic distance maps, and word clouds. The term scores for each parameter are visualised by sorting them in decreasing order. This term score visualisation greatly influences parameter identification. A two-dimensional intertopic distance map shows the parameters as parameter circles whose sizes correspond to the number of words used to describe the parameters in the dictionary. Word cloud is a graphical representation of words. Based on frequency and relevance, the famous words and phrases in the articles are highlighted through Word Cloud. It provides intuitive and quick graphical insight, which may lead to thorough analyses.

### 4.7. Validation and Visualization

Results can be validated internally and externally. Internal validation of a parameter involves investigating and discussing the posts related to the parameter. We discussed how we perceived the correlation between the documents and the parameters in most of the documents in our dataset. External validation is performed by comparing the knowledge and parameters gained from the analysis of the subject dataset with information from other datasets and information sources. For the visualisation, various visualisation methods are used for internal and external validation. Many visualisation methods are used to describe the datasets, the clusters of documents, and the parameters that have been discovered. These visualisations are created using several Python libraries, including LDAvis (Sievert and Shirley 2014), Plotly (2022), and Matplotlib (2022).

## 5. Parameter Discovery for Multi-Generational Labour Markets from LinkedIn

This section introduces and explains the parameters detected using our LDA model based on the LinkedIn dataset. These parameters are divided into five macro-parameters. Section 5.1 provides an overview of the parameters and macro-parameters. Section 5.2 gives the quantitative analysis of the parameters. The five macro-parameters are discussed in Sections 5.3–5.7.

### 5.1. Overview and Taxonomy

Using the LDA model, we detected 15 parameters in the LinkedIn dataset. We grouped these 15 parameters into 5 macro-parameters. We tried clustering the data with different number of clusters and decided to discover 15 clusters based on the coherence score (this will be discussed further in the next section) The LinkedIn dataset parameters and macro-parameters are listed in Table 2. In Column 1, the parameters are categorized into five macro-parameters: Generations-specific Issues, Skills and Qualifications, Employment Sectors, Consumer Industries, and Employment Issues. The second and third columns list the parameters and the cluster number, respectively. The fourth column shows the percentage of the keywords in each parameter. The fifth column displays the top keywords associated with each parameter.

**Table 2.** Parameters and macro-parameters of labour market from a multi-generational perspective.

| Macro-Parameter | Parameter | No | % | Keywords |
|---|---|---|---|---|
| Generations-specific Issues | Crimes and Racism | 10 | 3.3 | Woman, Cancer, Child, Black, Abuse, State, White, Right, Skin, Cell, Legal, Body, Hospital, Police, History, Court, Japan, American, Medical, Infant. |
| Skills and Qualifications | Leadership | 2 | 16.1 | Team, Community, Leadership, Opportunity, Event, Business, Support, Leaders, Student, Join, Future, School, Work, Programme, Great, People, Education, Young, Learn, Share. |
| | Skills and Qualifications | 13 | 2.2 | Selection, Criteria, Professional, Qualifications, Relevant, Position, Work, Experience, Internships, Language, Skills, Salary, Hiring, Recruiter, Organization, Excellent, Communication, Ability Plan, Task. |
| Business and Employment Sectors | Family Business | 3 | 12.8 | Business, Company, Team, Experience, Management, Data, Customer, Skill, Work, Technology, Need, Role, Product, Client, Working, Manager, Project, Opportunity, Industry, Digital. |
| | Remote Work | 7 | 3.5 | Work, Employee, Remote, Working, LinkedIn, Podcast, Workplace, Office, Company, Culture, Worker, Comment, Week, Employer, Great, Article, Episode, Video, Hybrid, Read. |
| | Recruitment | 8 | 3.3 | Recruiter, Hire, Candidate, Experience, Senior, Looking, Resume, Apply, Recruitment, Manager, please, Recruit, Position, Years, Engineer, Share, Interest, Comment, Job, Year. |
| | Entrepreneurship | 11 | 3.1 | Startup, Founder, Investor, Tech, Web, Entrepreneur, Venture, Investment, Estate, Real, Innovation, Funding, Fintech, Capital, Ecosystem. |
| Consumer Industries | Brand Marketing | 5 | 6.8 | Brand, Market, Marketing, Medium, Social, Content, Consumer, Price, Growth, Rate, Product, Inflation, Increase, Year, Audience, Economy, Business, Interest, High, Generation. |
| | Energy Sector | 9 | 3.3 | Project, Energy, Design, Construction, Water, Engineering, Power, Plant, Space, Manufacturing, Building, Area, Production, Safety, Engineer, System, Material, Industry, Vehicle, Site. |
| | Celebrations and Events Industry | 6 | 3.7 | India, Award, Century, Africa, International, Food, Innovation, President, Country, Congratulations, Principal, South, City, Year, Global, National, Global, July, Holiday, Employee, Special. |
| | Retirement Industry | 14 | 1.7 | Retirement, Tax, Pension, Years, Old, Income, Retire, Plan, Age, Planning, Money, Insurance, Family, People, Hospital, Care, Health, Medical, Support, Physical. |
| | Entertainment Industry | 12 | 2.6 | Music, Game, Sport, Film, Song, Artist, Army, Play, Veteran, Fashion, Design, Player, Star, Millennial, Collection, Winner, Movie, Feature, Service. |
| Employment Issues | Mental Health | 1 | 26.6 | People, Time, Make, Life, Know, Like, Want, Work, Take, Would, Many, Need, Years, Love, Even, Come, Start, Help, Back, Could. |
| | | 4 | 9.4 | Health, Care, Generation, People, Need, Population, Change, Woman, Child, Research, Mental, Aging, Boomer, Family, Increase, Impact, World, Community, Baby, Healthcare. |

A taxonomy of the multigenerational workforce is shown in Figure 2. The parameters and macro-parameters found on LinkedIn posts are used to create the taxonomy. The macro-parameters are displayed on the first level branches, and the discovered parameters are displayed on the second level branches.

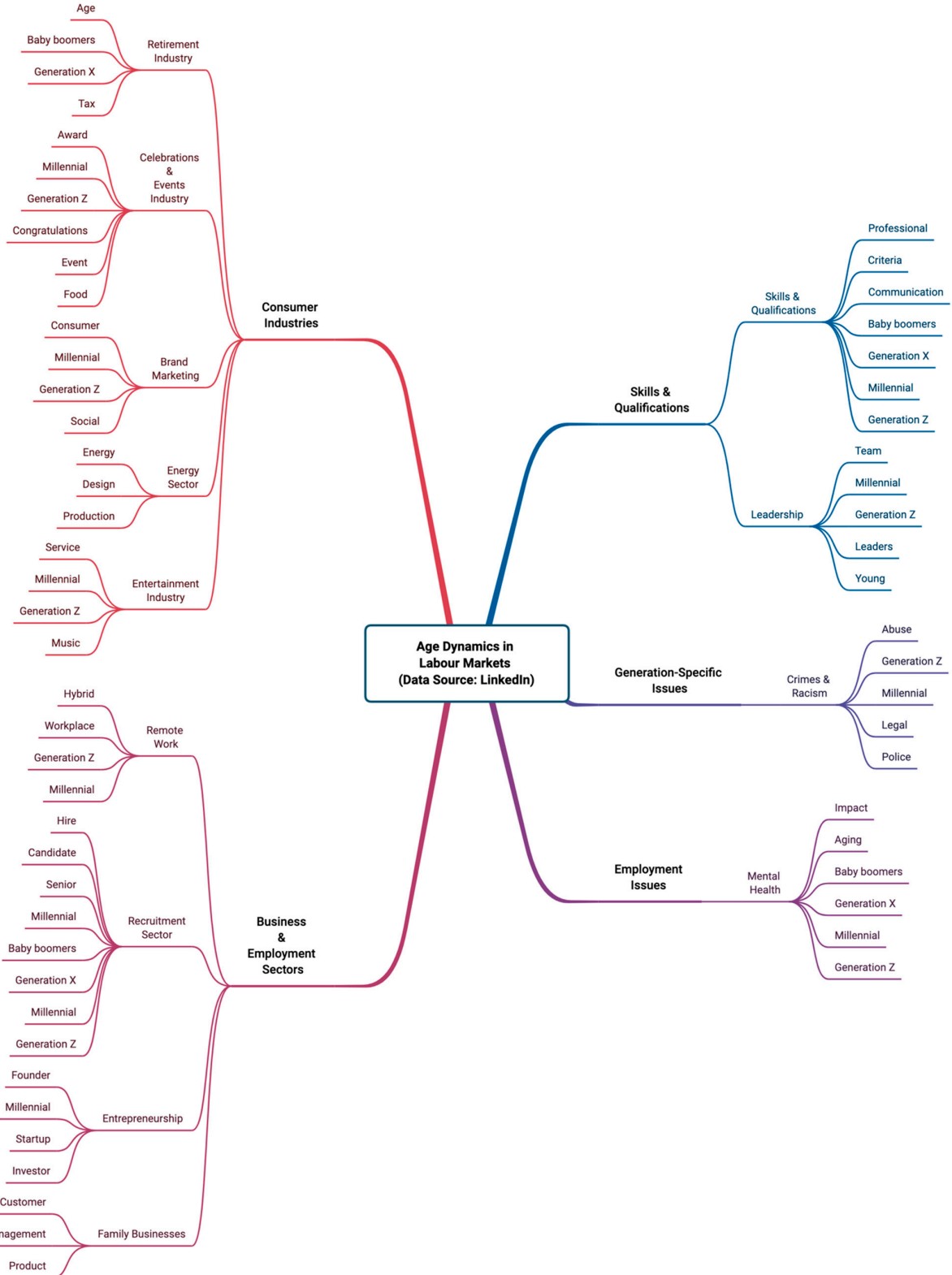

**Figure 2.** LinkedIn perspective taxonomy of the multi-generational labour market.

*5.2. Quantitative Analysis*

This Section discusses the coherence measures, intertopic distance map, and the number of posts on each topic. We have created the models using the Gensim LDA package. We have chosen different top models with the optimal number of topics based on the coherence measures. Table 2 presents the four different coherence measures employed to find the optimal number of topics. We have chosen the number of topics to be 15 since it has the best scores (highlighted in bold) in two different measures, as shown in Table 3.

**Table 3.** Coherence measures for selecting the optimal number of topics.

| Dataset | Num Topics | Coherence Value | U_mass Value | C_uci Value | C_npmi Value |
|---------|-----------|-----------------|--------------|-------------|--------------|
| LinkedIn posts | 10 | 0.538 | −3.52 | −0.23 | 0.049 |
| | 15 | 0.57 | −3.204 | **0.067** | **0.071** |
| | 20 | 0.56 | −3.939 | −0.493 | 0.047 |
| | 25 | **0.582** | −5.768 | −1.877 | −0.005 |
| | 30 | 0.558 | −5.252 | −1.494 | 0.008 |
| | 35 | 0.548 | −5.83 | −2.155 | −0.015 |
| | 40 | 0.521 | −7.29 | −3.364 | −0.077 |
| | 45 | 0.519 | −6.643 | −2.908 | −0.05 |
| | 50 | 0.511 | −7.119 | −3.424 | −0.076 |
| | 55 | 0.516 | **−7.796** | −3.975 | −0.097 |

Figure 3 shows the inter-topic distances and the most important words among the extracted 15 topics information. The intertopic distance map revealed the circles which represent the topics. The size of the circle is directly proportional to the topic's relevance. Generally, a good topic model will have significant, less overlapping circles throughout the chart. On the other hand, a poor topic model will have highly overlapping circles clustered in one quadrant.

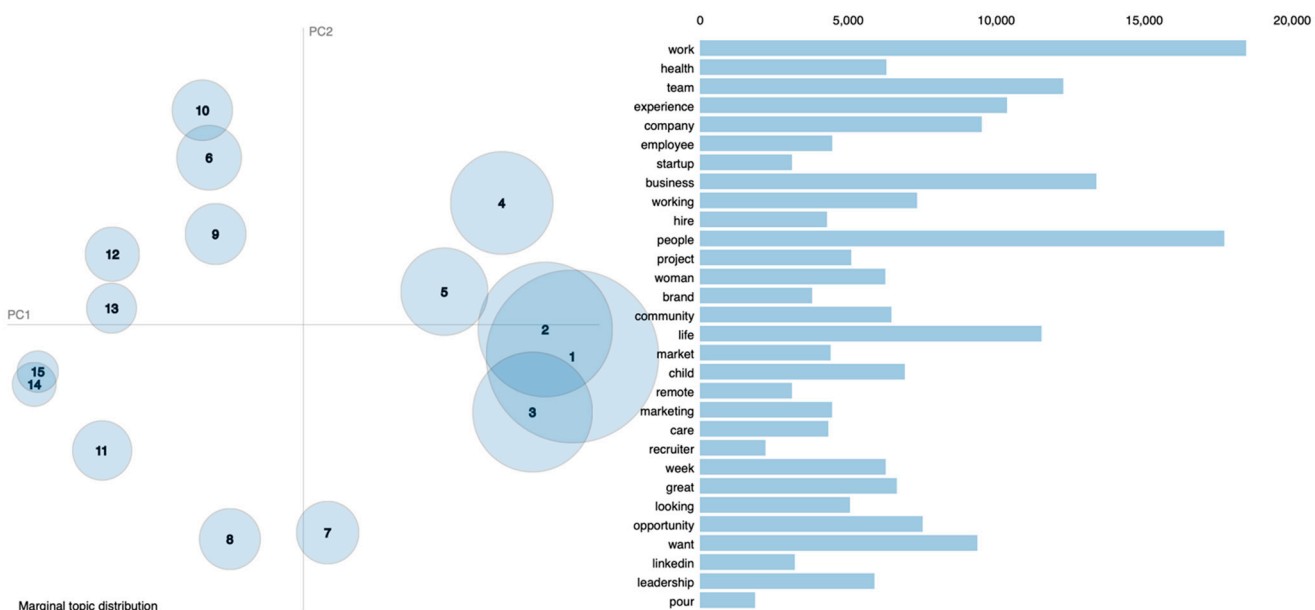

**Figure 3.** The intertopic distance map of the topics and the most important words.

Figure 4 shows the most relevant keywords that were detected by our LDA model in Topic 7 (Remote Work). The blue bars mean the overall term frequency while the red ones mean the estimated term frequency within the selected topic (Topic 7: Remote Work). Remote Work depicts the following keywords work, employee, remote, working, LinkedIn, podcast, workplace, office, company, culture, worker, comment, week, employer, great, article, episode, video, hybrid, and read.

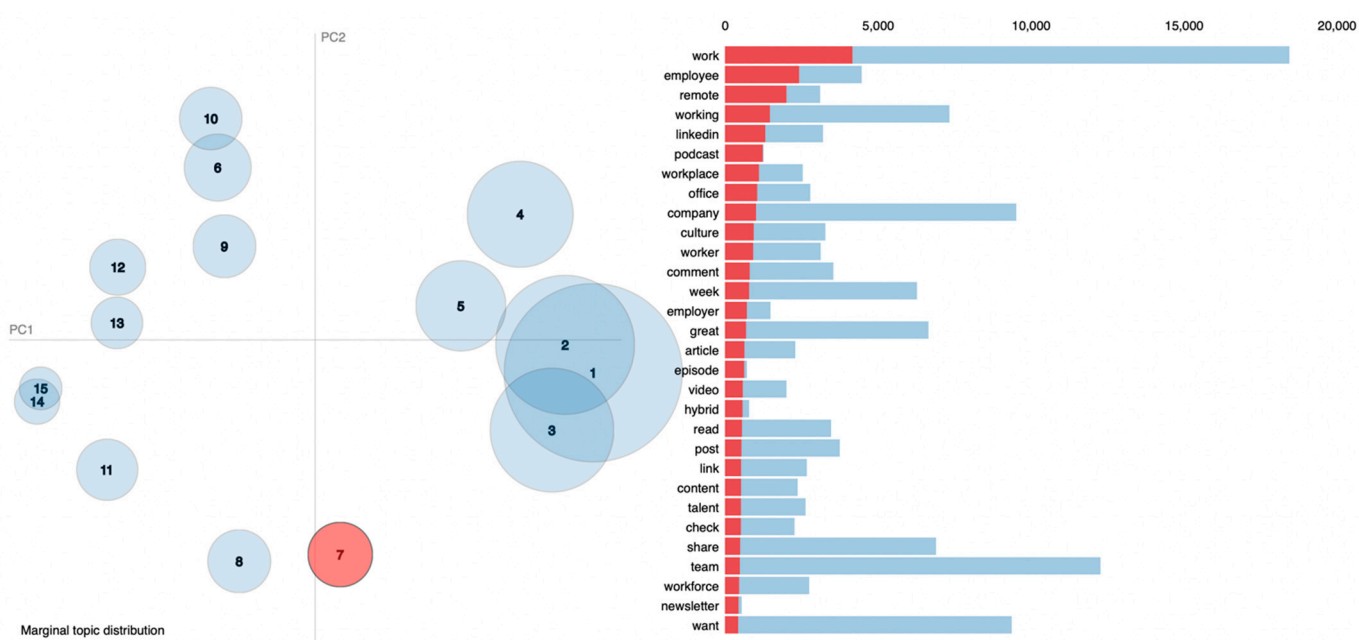

**Figure 4.** The top 30 relevant keywords in Topic 7 (Remote Work).

The histogram in Figure 5 shows the number of documents on each topic. The *x*-axis represents the number of a particular topic, while the *y*-axis depicts the number of documents on that topic. We have removed Topic 15 from the analysis since the majority of the posts in it are in different languages.

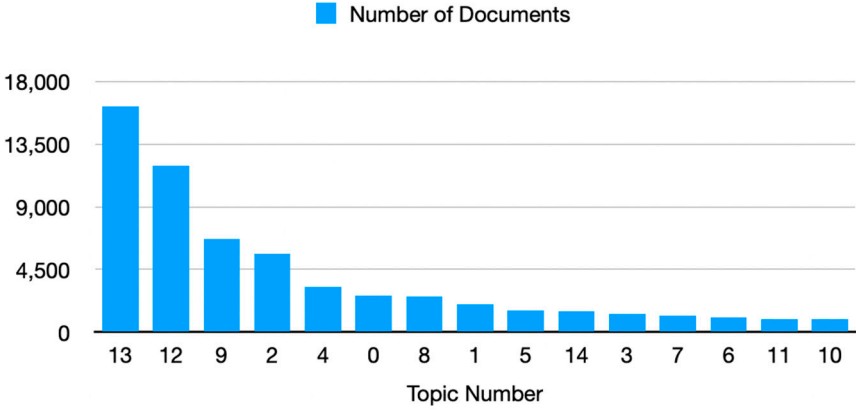

**Figure 5.** Number of posts in each topic (Source: LinkedIn).

Figure 6 shows the ten most important keywords for each parameter. Vertical lines indicate parameter keywords, and horizontal lines indicate importance score. The lightest blue color means the most important keyword while the darkest blue color means the least important.

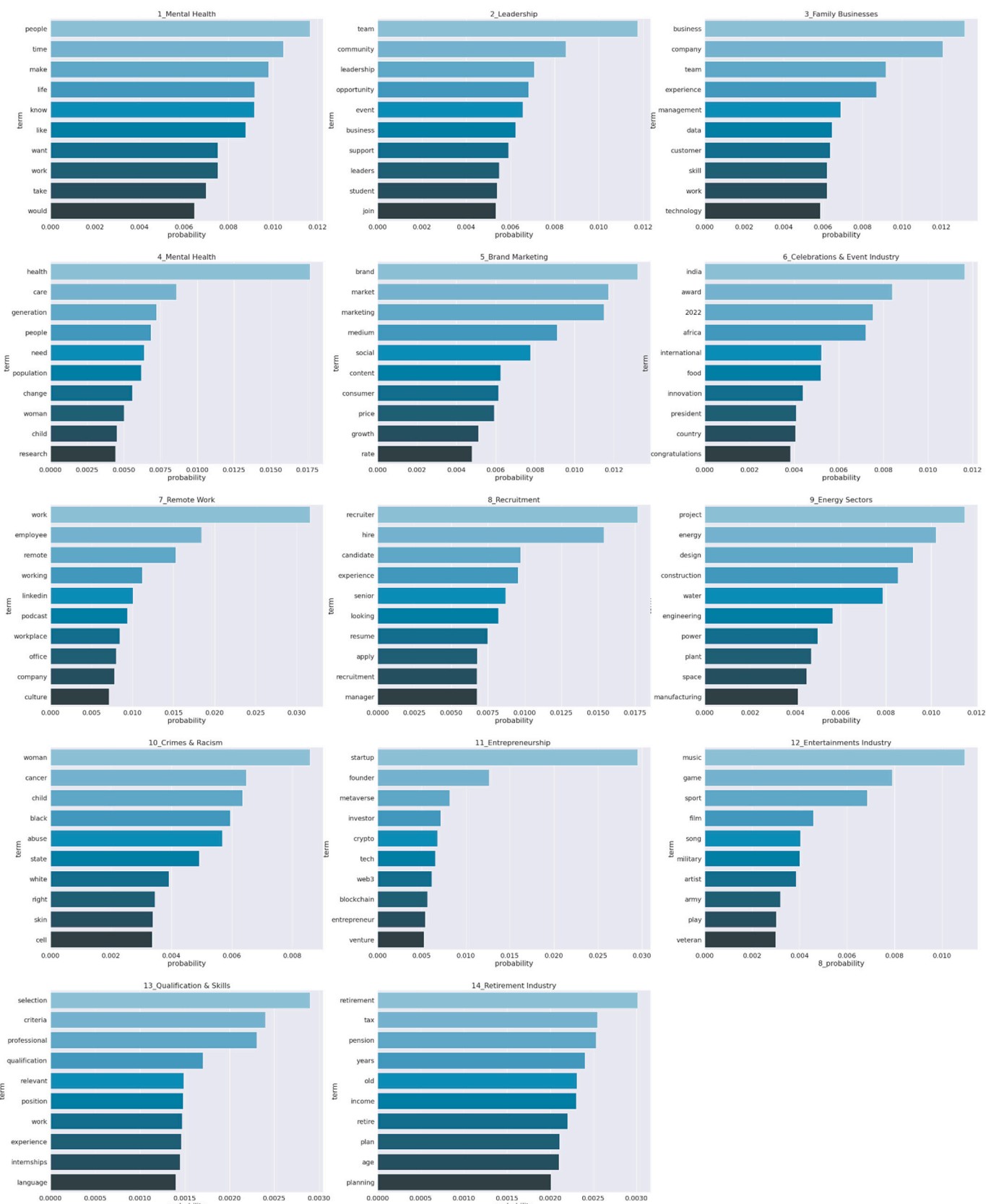

**Figure 6.** Term score.

*5.3. Generation-Specific Issues*

This macro-parameter has only one parameter; however, analysis of additional datasets in the future would likely discover additional parameters.

Crimes and Racism

The parameter Crimes and Racism contains keywords such as woman, child, black, abuse, state, white, right, skin, cell, legal, body, hospital, police, history, court, Japan, American, medical, and infant. Unemployed young people are often the victims of crimes, such as robberies and assaults. In some cases, young people are even the perpetrators of crimes. In other cases, young people may be homeless or have been evicted from their homes because they could not pay their rent or other bills.

This parameter admits a direct correlation between unemployed young people and crimes. As an example, this issue was discussed in the post "*The United States is home to more than four million , youth, young people ages 16 to 24 who are neither in education nor employment. These individuals are five times more likely to have a criminal record than their peers . . .* "(RISE 2022). Moreover, there is also a relationship between unemployment and anxiety and depression which leads to crime. The following post refers to this issue "*Long term unemployment, crime, family breakdown or homelessness usually originate from or relate to struggles with health, with anything from alcoholism to serious anxiety and depression . . . I advocated our objectives of ensuring we prevent crime and enable rehabilitation through more integrated mental health, substance misuse, help with learning needs and other support, so we can tackle the causes of crime.*" (Thomas 2022).

This parameter also includes posts about racism at work which is the act of discrimination against someone based on their race. It can take many forms, including but not limited to, inappropriate racial jokes, cyberbullying, unconscious bias and prejudice, and unfair hiring practices. For instance, the following post, " *. . . ! The RARE (Racism and Racial experiences) workgroup at @Rice University has come up with this Emotional Labor Invoice . . . The next time someone micro-aggresses you based on your gender, orientation, race–feel free to drop them this invoice . . .* " (Harry 2022).

*5.4. Skills and Qualifications*

This macro-parameter comprises two parameters.

5.4.1. Leadership

The parameter Leadership includes keywords such as team, community, leadership, opportunity, event, business, support, leaders, student, join, future, school, work, programme, great, people, education, young, learn, and share. Leadership skills are the ability to influence others. They are a key part of success in business, and they are important for any successful person to have. One of the most important skills for a leader is being able to effectively communicate with others. That means being able to listen and understand what they are saying, and how they are feeling. For instance, a post mentioned, "*The leader's —job is to create such a space and make their employees feel free to express their opinions and ideas.*". It also means being able to make decisions quickly and effectively, so that everyone feels heard and respected. Thus, our millennials and younger generations need to learn leadership skills before they supposedly "need" them. To learn about leadership, we need not wait until we become supervisors or managers. Teenagers and even younger people can be leaders. Therefore, school curriculums should include lessons on leadership and personal growth (Ong 2022).

The LinkedIn community also discussed opportunities to learn new skills through participation in workshops, team-building activities, or other programmes. For instance, a post mentioned, "*On today's episode of Being [at Work], Marketing Intern, Audrey Caron, provides a daily dose of leadership through the lens of Gen-Z.*".

Multigeneration experiences through better leadership and management can create new income streams for the organisation. By creating a diverse group of workers trained

to be comfortable speaking in front of groups of people, first-time reality participants can become valuable contributors to the community through their leadership capabilities and experiences.

### 5.4.2. Skills and Qualifications

The parameter Skills and Qualifications includes keywords such as selection, criteria, professional, qualifications, relevant, position, work, experience, internships, language, skills, salary, hiring, recruiter, organisation, excellent, communication, ability plan, and task. The job skills and qualifications are used to identify specific job skills that are required to perform certain jobs. Skills are one of the most important factors for jobs. One of the most important skills for a job is being able to communicate with others in a way that is understandable and consistent across different generations. This is particularly important if the workplace has a multigenerational team environment where there are many people who may have different ages, skills, or experience. In addition, skills are very important for the young generations and for the future of the countries. The Education Fit for the Future in Wales report stated that " ... *education and qualifications are crucial for different jobs in the future* ... " (Jones 2019). In addition, a post stated, *"Catherine Cain, Assistant Vice Principal and Careers Lead at All Saints' Academy, Cheltenham explains how it is a priority for their curriculum and future education pathways to match the needs and skills required for the upcoming development".* Furthermore, nowadays, programess to provide skills to young generations become available as has been described: " ... *Employment and skills for development in Africa (#E4D) have partnered to establish a landmark program to provide skills to young generations in technical training in renewable energy"* (WTS Energy 2022).

### 5.5. Business and Employment Sectors

### 5.5.1. Remote Work

The parameter Remote Work depicts the following keywords, work, employee, remote, working, LinkedIn, podcast, workplace, office, company, culture, worker, comment, week, employer, great, article, episode, video, hybrid, and read. Remote Work discusses how the workplace is changing. Many people are starting to embrace a "hybrid" or "remote" working model. This means working from home some days while also spending time in the office on other days. Others are choosing to work part-time or freelance. For instance, working and commuting from home has become a viable option for millennials and Gen Z. This is because technology is making the workday more flexible than ever. For example, someone mentioned, " ... *This is the first fully remote company I have worked for, and this next chapter in my career is exciting* ... ". Moreover, according to a survey by Owl Labs (Labs 2020), *"Remote employees save an average of 40 minutes daily from commuting. Since 2020 people have been meeting by video calls 50% more since COVID-19. After COVID-19 92% of people surveyed expect to work from home at least 1 day per week and 80% expected to work at least 3 days from home per week. 23% of those surveyed would take a 10% pay cut to work from home permanently. People are saving on average close to 500 dollars per month being at home during COVID-19. Resulting in savings close to $6000 per year. A mere 20–25% of companies are paying some of the cost for home office equipment and furnishings. 81% of those surveyed believe their employer will continue to support remote work after COVID-19. 59% of respondents said they would be more likely to choose an employer who offered remote work compared to those who didn't* ... ".*

### 5.5.2. Recruitment Sector

The parameter Recruitment includes keywords such as recruiter, hire, candidate, experience, senior, looking, resume, apply, recruitment, manager, please, recruit, position, years, engineer, share, interest, comment, job, and year. The recruiting world is constantly changing. Recruiters are required to possess more skills than ever before. To succeed in this competitive market, recruiters must be able to adapt and be flexible. One of the essential skills a recruiter can possess is interacting effectively with multigenerational candidates.

At the same time, the critical skills for multigenerational candidates are passion, teamwork, productivity, and commitment. Many different types of candidates make up the multi-generational workforce, including Gen Z, Millennials, Xers, and baby boomers. These candidates come from various backgrounds, but they have one thing in common: a desire to work in a team environment that allows them to be flexible and adaptable. For instance, someone mentioned the following issue: "*My role is focused on recruiting the next generation. However, I truly believe in a multi-generation workforce.*". Furthermore, a report from the Organization for Economic Co-operation and Development (OECD) on Promoting an Age-Inclusive Workforce (OECD 2020) showcased how age inclusion can provide a competitive advantage and mentioned, "*The research projects that multi-generational workforces will be more efficient and productive, leading to a more profitable economy by raising per capita GDP by 19% over the next three decades . . . *".

### 5.5.3. Family Businesses

The parameter Family Business includes many keywords such as business, company, team, experience, management, data, and customer. The parameter includes posts about many types of businesses. Some of them are small, large, home-based, family, and franchise businesses. The most discussed type of business is family businesses. Successful, long-term family businesses are "generative" because they renew continuous luminosity across generations and have a good plan to survive. A post mentions, "*A very insightful article that highlights the importance of succession planning in family business . . . Transfer Your Family Business To The Next Generation With A Written Succession Plan*".

Multigenerational families often initiate shared educational and philanthropic activities. Each family has a unique way of giving back to the community that reflects their family's values and history. Family business histories and stories have been mentioned, such as "*There were so many great anecdotes to this family business story I couldn't fit them all in . . . *" (Roy 2022).

National Family Business Day has been mentioned too, which is a global movement that aims to raise awareness about the importance of inter-generational family in business and the importance of supporting family-run businesses. The day is celebrated on the first of January each year. A post mentions "*As four sisters running a successful inter-generational family business, we are proudly celebrating National Family Business Day 2021! Established in 1946 . . . *"(Towbars 2022). Another post states, " *. . . a Multi-Generation Family Business . . . learn how the family has grown, and continues to grow . . . *" (Iron 2022).

### 5.5.4. Entrepreneurship

The parameter Entrepreneurship contains the following keywords: startup, founder, investor, tech, web, entrepreneur, venture, investment, estate, real, innovation, funding, fintech, capital, and ecosystem. Entrepreneurship is a broad term describing the act of starting a business. It can refer to starting a small business on one's own, or it can refer to creating an innovative company with an idea that potential investors are willing to back. Entrepreneurship is often related to innovation, because it requires coming up with new ideas and solutions for existing problems. There are many different posts of entrepreneurship, including tech entrepreneurship, startup entrepreneurship, and venture entrepreneurship. Each type has its own set of requirements and challenges, but all entrepreneurs have one thing in common: a willingness to take risks. Entrepreneurs are also called "founders". The Kauffman Foundation and Vivek Wadhwa (Meyers 2020) found that the average age of successful founders is 41.9. A post mentioned this report, " *. . . Researchers found that the average age of a business founder in the United States is 41.9 years old. They discovered that older founders consistently had higher probabilities of success, at least until the age of sixty. A sixty-year-old start-up founder has a roughly three times higher chance of creating a valuable business than a thirty-year-old start-up founder . . . *".

Another issue that has been discussed is that talent is more important to some millennials than position. For instance, according to a recent survey conducted by the National

Venture Capital Association ([NVCA 2019](#)), the number of young people who are investing in venture capital has increased significantly in recent years. While many young people are not yet full-fledged entrepreneurs, they are beginning to see the benefits of starting their own businesses. The NVCA survey found that more than half of millennials (48%) are considering investing in their own business, compared to only 35% of baby boomers. While many people may not have the financial resources to start their own business, it is still a great way for them to start. Another issue that has been discussed is that talent is more important to some millennials than business.

On the other hand, baby boomers also have their own businesses. A post mentions, "250 k Baby Boomers own businesses . . . what is going to happen when they want to retire? . . . This provides a massive opportunity for young entrepreneurs" ([Shiver 2022](#)).

*5.6. Consumer Industries*

5.6.1. Brand Marketing

The parameter Brand Marketing is characterised by brand, market, marketing, medium, social, content, consumer, price, growth, rate, product, inflation, increase, year, audience, economy, business, interest, high, and generation. Digital marketing is the process of using digital platforms such as social media, blogs, and websites to communicate with customers and generate leads. The key to success for digital marketing for brands is to understand the needs of the target audience and create a brand that resonates with them. This includes understanding their demographics, interests, and likes/dislikes. Moreover, digital marketing can be a powerful tool for brands because it allows them to reach out to new audiences and engage with them through social media. By building relationships with new users, brands can build a following that drives more sales and growth. In addition, digital marketing can also help in keeping track of the customers' progress on social media platforms like Facebook, Twitter, or Instagram. This gives businesses the opportunity to respond quickly if they need more information or updates about their products or services. Furthermore, digital marketing for the brands is the key to reaching Generation Z and future generations. These generations comprise some of the most voracious social media users. For example, someone mentioned, " . . . *Physical advertising by billboards and banners never had the right appeal for generation Z. To market a product or service to them, digital marketing is the key. Brand marketing through platforms like Instagram, public forums, and chat groups is the new marketing strategy that works.*" ([Gryffindor 2022](#)).

5.6.2. Retirement Industry

The parameter Retirement contains keywords such as retirement, tax, pension, years, income, retire, plan, age, planning, and money. This parameter includes discussions on retirement issues and planning, which involves assessing the current financial situation, assessing future needs (financial, healthcare, etc.), and planning for an appropriate level of savings to support one's post-work life. It also discussed the baby boomers' situation. For instance, the number of adults over 65 will exceed the number of children in the country for the first time in 2023, according to the Bureau of Labor Statistics ([U.S. Bureau of Labor Statistics 2022](#)). Some other baby boomers' retirement issues are mentioned below.

> *"Baby Boomers are moving into retirement and taking their hard-earned expertise with them, while younger workers are reluctant to even consider entering the field service industry. Those that are entering the industry face a growing skills gap that impacts costs and service . . . ".* ([DeepHow 2022](#))

> *"The Boomer population is approaching retirement! You know what that means, the long-term care industry will be larger than it's ever been. How can an already short, staffed industry provide for a growing market? Here are a few tips addressed to healthcare providers to best prepare for and effectively service this demographic . . . ".* ([Leigeber 2022](#))

> *"It's time to clear the boomers for takeoff . . . For any industry to thrive, it must value its people of all ages and generations. Beyond the friendly skies, pilots need an age-friendly*

*industry: one that continues to mandate safety while still valuing age and experience, retaining those pilots who have reached 65 but have still been found to be healthy enough to fly while parting ways with those who have not . . . ".* (Applewhite 2022)

*"The baby boomers' great retirement has been forecasted for years and comes as no surprise. What makes it complicated is that the people retiring have years of experience and are typically in senior-level positions that can't be filled by new recruits. If you have a position that requires 15 years of experience, the only way to fill it is to give 15 years of experience. There's a problem that's been developing for years,".* (Omar 2022)

### 5.6.3. Energy Sector

The parameter Energy Sector depicts keywords such as project, energy, design, construction, water, engineering, power, plant, space, manufacturing, building, area, production, safety, engineer, system, material, industry, vehicle, and site. The parameter is used to describe electricity production using a variety of technologies. Some technologies, such as steam boilers, are over 100 years old, while others, like wind turbines, are newer. There are also posts about hiring for power generation jobs. Below are some examples of posts that are related to Energy Sectors.

*"Dear All, We are hiring for the below-mentioned positions with us, Apollo Power Systems is a 30-year-old company in the business of Power Generation . . . ".* (Kumar 2022)

*"First, two facts:1. By 2030, 58.7 GW of renewables are expected to be connected to the Saudi power grid. That is approx. 50% of total installed generation capacity by then . . . ".* (Al-Awami 2022)

### 5.6.4. Celebrations and Events Industry

The Celebration parameter includes keywords detected by our LDA model such as award, international, food, innovation, president, country, congratulations, principal, south, city, year, global, national, global, and July. People share celebration posts about what they have learned with the other team or the community. Generally, these posts (Rafalski 2016; Canopius Group 2022) let everyone know about their work and achievements over the years, whether it is a new customer acquisition strategy, a product launch, or a new job promotion. Moreover, companies have a history of working with special employees and their families. For example, a company has worked with its team members to celebrate birthdays and work anniversaries. This is done by providing them with special items such as gift cards and other perks. Some companies also offer their employees a day off from work to celebrate birthdays and holidays. The company also provides employees with a day off from work during the month. Sometimes, they are offering them a free meal and other perks such as free drinks and free food at the end of the day. We list some of the posts below.

*"On Friday I had the privilege of attending the 25-year anniversary party . . . Although the focus was on celebration there were a few things that really resonated with me from the stories shared:—the positive impact from parents who have a strong work and family ethic and how this shapes kids from a young age- the beauty that is a business that has multi-generations of family working within it, the pride and drive that comes from a shared goal . . . ".* (Elouise 2022)

*" . . . She is an instrumental part of our Benefits Eligibility & Enrollment Systems (BEES) Team, helping our clients with their eligibility needs and assisting with the training and development of the newer members of our team. She is a true team player, and we are SO lucky to have her on OUR team! Congratulations on your promotion and thanks for 8 great years . . . ".* (Catto 2022)

### 5.6.5. Entertainment Industry

The parameter Entertainment (Industry) includes keywords such as music, game, sport, film, song, millennial, artist, army, play, veteran, fashion, design, player, star, col-

lection, winner, movie, feature, and service. This parameter is related to discussions on different topics in media such as music, films, games, and movies. As an example, over the years all music has undergone tremendous changes. Thus, each generation has several songs that define it. Furthermore, a post mentions that millennials and Gen Z listen to music more than other generations *"According to data that Luminate shared with Axios. State of play: Americans born in the '90s and 2000s are listening to music from the decade they were born at higher rates than other generations listen to music from their birth decades, TikTok plays a role here. And those born in the '60s—'80s, have increased their music consumption over the past couple of years as the pandemic drove an increase in streaming . . . "* (Luminate 2022). In addition, music is a major expense for millennials and Generation Z. For instance, the Music in the Air (Ingham 2022) report discussed *"Millennials and Generation Z are spending more of their money on music than other age groups. I often say people put as much emotional value on music as just about anything and given that it remains highly undermonetized. Spending is 40% below its historical peak, but consumption grows every year. Emerging platforms (social, video games, podcasts) now represent almost 1/3 of the record industry's ad funded revenues and should grow to 40% by 2030 if not sooner . . . "*. On the other hand, as a platform for influencer marketing, TikTok has quickly built a strong reputation between millennials and Gen Z. For example, someone discussed and posted *"Has the time come for me to be active on TikTok? It is clearly a dominant force in social media and marketing . . . An interesting discussion on how the ever-changing social media landscape is producing new roles and subsequent skill needs being filled by millennials."* (Wilson 2022).

*5.7. Employment Issues*

This macro-parameter has only one parameter; however, analysis of additional datasets in the future would likely discover additional parameters.

Mental Health

The parameter Mental Health includes keywords such as health, care, generation, people, need, population, change, woman, child, research, mental, ageing, boomer, family, increase, impact, world, community, baby, and healthcare. Mental health is a state of well-being in which an individual feels comfortable and confident. It can be affected by many factors, such as genetics, environment, age, and life experiences. The parameter has shown that mental health can be affected by various factors throughout the life cycle. For instance, how COVID affected our mental health (OX 2022). The most recognised stages of mental health are childhood, adolescence, and adulthood (Sharma 2022). However, the mental health of individuals can change throughout their lifetime. For instance, it has been mentioned that *"Young employees feel disconnected at work and home, and according to LinkedIn, 66% of Gen Z employees want a company culture built on mental health and wellness. Likewise, millennials have been pushing for better mental health benefits since joining the workforce . . . "* (Antonio 2022). In addition, someone mentioned podcasts such as *"Fitness for the Brain | On this edition of the Boomers Today podcast . . . "* which discuss how to combine brain training and physical training for better mental health after retirement (Metro 2022).

## 6. Discussion

Recent research on viral social networks, such as Twitter and Facebook, was intensive, and datasets are readily available. However, LinkedIndoes not have a dataset gathering information (posts) perhaps because it is a design decision by the company or it is mainly for professionals and so it is not expected to grow as big as other social networks for journal public. Our study relies on LinkedIn as the data source for our study due to the nature of the study which focuses on labour markets involving professionals. The dataset was obtained using web scraping and social mining techniques.

In this paper, we applied our deep journalism approach to LinkedIn to automatically extract and analyse big data to provide the public with information about labour markets; people's skills and education; and businesses and industries from multi-generational per-

spectives. The case study focuses on a multigenerational workforce using LinkedIn posts. Specifically, we discovered 14 parameters using machine learning from LinkedIn posts using LDA algorithms and grouped them into 5 macro-parameters, namely, Generations-specific Issues, Skills and Qualifications, Employment Sectors, Consumer Industries, and Employment Issues. We developed a software tool from scratch for this work that implements a complete machine learning pipeline using the dataset.

The parameters discovered through AI-based journalism applied to LinkedIn media focused on many issues in labour markets. For instance, despite the benefits that remote work offers, it is important for millennials and future generations to understand the challenges that come with working from home and how to overcome them. One of the biggest challenges that working from home can pose is an increased risk of burnout. The nature of working in an office environment allows time for employees to decompress and recharge after a long day at work. Working from home does not offer this luxury, and as a result, employees often feel more pressure to be productive all the time. This can lead to burnout and high levels of stress, which can negatively impact health and wellbeing. To combat this, it is important to take breaks throughout the day. This will help to keep employees feeling refreshed and ready for another productive day at work. Furthermore, it is important to take care of mental health because it can affect the physical health as well as the relationships with other people. Adults and seniors should feel comfortable talking about their feelings with their friends and family members. Figure 7 shows a taxonomy of age dynamics in labour markets discovered by our tool.

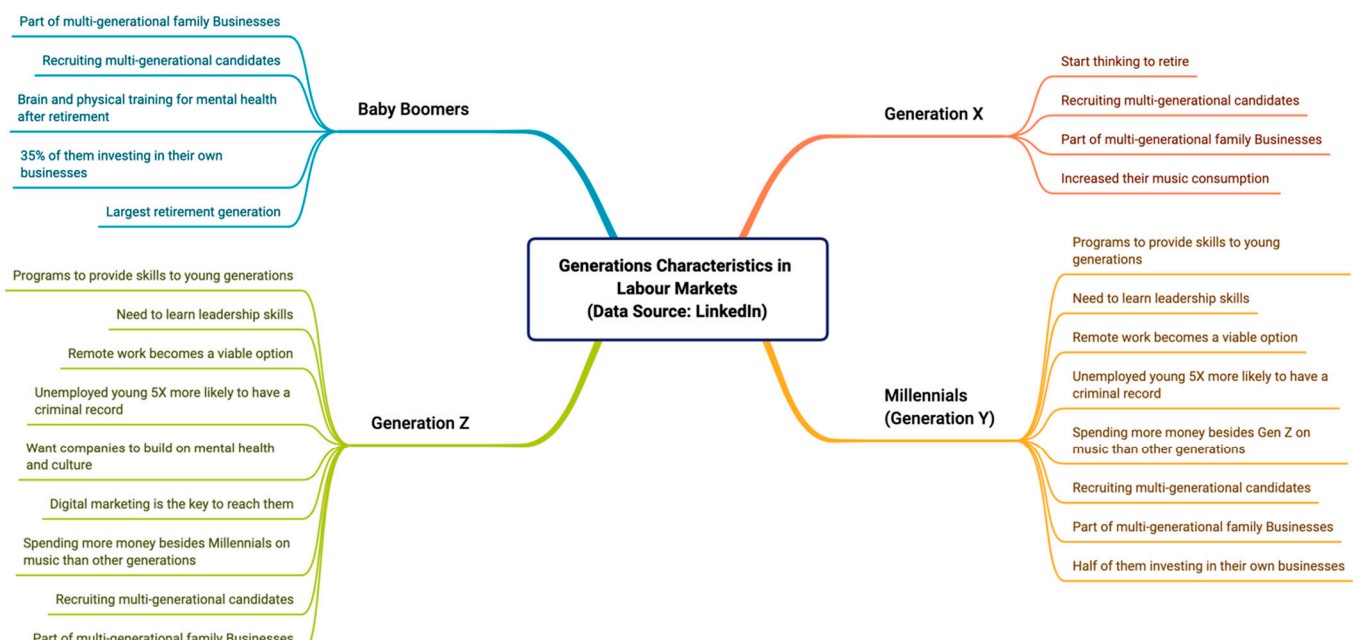

**Figure 7.** Different Generations in Labour Markets and Their Characteristics.

Another important issue that we discovered and discussed relates to hiring and retaining employees, which is becoming more difficult for organisations nowadays. While organisations can manage their workforce effectively, they may face challenges in hiring and retaining the right employees for their organisation. For example, multi-generational employees may be more likely to leave the organisation because of a lack of opportunities for advancement. Furthermore, it can be challenging to create a culture that is suited for all generations. To help with this situation, organisations may need to adjust their policies so that they are more inclusive of different generations in the workplace. This could include creating flexible work schedules, offering training programmes that are relevant to each employee's needs, and ensuring that communication is open between different generations.

We also discovered that baby boomers in their 50s and 60s are now retiring, and by 2023, there will be more retirees than children in the United States. This will have a huge impact on the workforce. A shrinking labour pool will drive the demand for employees higher, causing the cost of hiring and training a new worker to increase and the number of qualified applicants to decline. One solution is to bring in more retirees and offer them flexible, part-time scheduling options.

Additionally, we found that unemployed Youth (Gen Y and Gen Z) in America are five times more likely to have a criminal record. Moreover, racism issues occuring at work can have a serious impact on both the victim and the workplace. Victims may feel unwelcome or uncomfortable in their jobs, and they may lose respect for colleagues or coworkers. In addition, a racist environment can cause employees to distrust one another and discourage teamwork. In addition, racist attitudes may lead to lower productivity and turnover among employees.

Businesses are well-positioned to remain competitive in the 21st-century economy. This is because there is no specific age to become an entrepreneur. At any age, entrepreneurs can start small businesses on their own or work for established businesses. They can also look for ways to invest in startups or become involved in other types of entrepreneurial activities. However, entrepreneurs must be prepared for the many challenges that lie ahead. Furthermore, from a brand marketing perspective of the business, it was found that the new marketing theory adopted by digital marketing relies on modern digital technology and media to meet the needs of youth customers (Gen Y and Gen Z).

## 7. Conclusions

The social and economic situation around the world has agonized people. COVID-19, environmental disasters, geopolitical situations, and wars have exacerbated the economic and social situations throughout the world. The Great Resignation and Quiet Quitting phenomena coupled with rapidly changing generational attitudes are bringing unprecedented and uncertain changes to labour markets. In this paper, we introduced our data-driven deep journalism approach to LinkedIn media and applied it to discover age dynamics in multi-generational labour markets. We created a software tool for discovering parameters for age dynamics in labour markets using LinkedIn data. We collected a total of 57,000 posts from LinkedIn and used it to discover 15 parameters by LDA and grouped them into 5 macro-parameters, namely Generations-Specific Issues, Skills and Qualifications, Employment Sectors, Consumer Industries, and Employment Issues. We described the proposed approach and tool and discussed the discovered parameters in detail.

This work is part of our work on Deep Journalism, introduced by (Mehmood 2022; Ahmad et al. 2022), which uses machine and deep learning to gather and produce information and knowledge, design, and other parameters to provide a holistic and multi-perspective view of a sector and help bridge the knowledge and collaboration gaps that exist to reduce inefficiencies and failures. The deep journalism approach used in this paper can provide everyone with objective, cross-sectional, and multi-perspective information. It can improve the rigor of journalism by making it simple to generate information using machine and deep learning, and it can make tools and information available so that anyone can discover information about issues of public concern. The broad purpose of this work is to understand various topics of public concern and use this understanding and knowledge to create awareness of important issues and drive future research on this topic using cutting-edge technologies.

The knowledge and parameters discovered in this paper can be used to develop a better understanding of labour markets and labour economics, allowing academics, governments, industry, social enterprises, and other organisations to develop better theories and practices (policies, technologies, solutions, and industries), leading to stronger economies and labour markets, which in turn will strengthen societies around the world by embedding equality, sustainability, resilience, and opportunity.

In the future, we plan to extend the breadth and depth of this work with advanced deep-learning methods applied to different journalism problems, sectors, and media.

**Author Contributions:** Conceptualization, A.A.A. and R.M.; methodology, A.A.A. and R.M.; software, A.A.A.; validation, A.A.A. and R.M.; formal analysis, A.A.A. and R.M.; investigation, A.A.A., R.M. and F.A.; resources, R.M. and F.A.; data curation, A.A.A.; writing—original draft preparation, A.A.A. and R.M.; writing—review and editing, R.M. and F.A.; visualization, A.A.A.; supervision, R.M. and F.A.; project administration, R.M.; funding acquisition, R.M. All authors have read and agreed to the published version of the manuscript.

**Funding:** The authors acknowledge with thanks the technical and financial support from the Deanship of Scientific Research (DSR) at the King Abdulaziz University (KAU), Jeddah, Saudi Arabia, under Grant No. RG-11-611-38.

**Informed Consent Statement:** Not applicable.

**Data Availability Statement:** The data used for this work can be made available to the public subject to data terms and conditions.

**Acknowledgments:** The work carried out in this paper is supported by the HPC Center at the King Abdulaziz University.

**Conflicts of Interest:** The authors declare no conflict of interest.

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
