# Peer review of "Data-Driven Deep Journalism to Discover Age Dynamics in Multi-Generational Labour Markets from LinkedIn Media"

_journalmedia, doi:10.3390/journalmedia4010010_

Round 1

Reviewer 1 Report

Thanks for the opportunity to read this interesting manuscript. It proposed a machine learning pipeline to classify LinkedIn posts regarding ages in the labor market. The methods used in this study are clearly explained. However, this paper may not be a good fit for the journal. It reads more like a paper from computer science or information science. It claims that it uses a journalism approach, which I did not see in the manuscript (also, I don’t really know what a “journalism approach” is, and the author(s) did not discuss it). The first section, “The State of Journalism,” reads irrelevant from the rest of the paper. Moreover, this study is purely descriptive; the classification is not based on any theories, and after the classification, the author(s) did not analyze how these clusters associate with other variables (e.g., how the proportions of these clusters change over time), limiting its contribution to the field. Finally, as a methodology paper, the LDA used in this study is not novel.

Reviewer 2 Report

Excellent work in taking on a new way of analysis -- in a journalistic way. You might make that "use to journalism" more obvious in the paper. This is something the industry needs, but it also needs to be approachable.

Reviewer 3 Report

The scope of the paper is around “core function of journalism” with the focus on providing access to unbiased information. Authors have selected deep journalism and with data-driven Artificial Intelligence (AI) based journalism approach they studied how the LinkedIn media could be useful for journalism. They have used real data by collecting 57,000 posts from Linkedin.  They have used Latent Dirichlet Allocation algorithm (LDA) and grouped 15 parameters into 5 categories. The categories defined the scope of their research. The authors claim that this approach is not used before and also suggested future work that can be carried out. I would like to congratulate the authors for their work. They need to consider the following

1. Introduction section, although well-presented, is lengthy.

2. Create a new section after literature review and move section 1.4 “This work” into the new section. This will highlight the novelty of this research work.

3. The conclusions drawn from “Literature review” which is currently in section 1.4 should be moved to end of section 2 after section 2.1.  

4. Section 2 “Literature review” is short. Although the authors have reviewed similar research work, but it is spread in the manuscript and is in different sections. Due to this it gives the impression that limited literature is reviewed. It is recommended to move all such text into section 2.

5. The direction of arrow (Search query) in Figure 1” Data collection and storage block” doesn’t seem right. Is this showing “response to search query” or "search query" itself.

6. Provide reference to the “experts in the field” text as mentioned in section 3.2.

7. Move table 1 after section 3.2. Currently the page is empty. Check for other formatting issues as well.

8. The authors presented the implementation in “Methodology” section. The implementation work should be in a separate section.

9. A better flow will have section 1 “introduction”, section 2 “Literature review”, section 3 “Methodology”, section 4 “Implementation or similar heading”, section 5 “Validation/ results or similar heading”, section 6 “Discussion” and finally section 7 “Conclusion”. The authors have most of the information already in the manuscript, but the information is not in the relevant sections.

10. There are some formatting errors, brackets missing, etc. so further careful review is required to remove these errors.

11. Some references are missing like (xxx) in sections “This work” and “conclusion” sections. 

Round 2

Reviewer 3 Report

The authors have implemented the suggested changes. I am satisfied with their effort. I suggest a final minor English language check and sorting of any remaining formatting issues be carried and then the paper is ready to be published.